# Modulating oxygen coverage of Ti$_3$C$_2$T$_x$ MXenes to boost catalytic activity for HCOOH dehydrogenation

Tingting Hou[1,6], Qiquan Luo [2,3,6], Qi Li[1,6], Hualu Zu[1], Peixin Cui [4], Siwei Chen[1], Yue Lin [3✉], Jiajia Chen[3], Xusheng Zheng[3], Wenkun Zhu [5], Shuquan Liang[1], Jinlong Yang [3✉] & Liangbing Wang[1✉]

As a promising hydrogen carrier, formic acid (HCOOH) is renewable, safe and nontoxic. Although noble-metal-based catalysts have exhibited excellent activity in HCOOH dehydrogenation, developing non-noble-metal heterogeneous catalysts with high efficiency remains a great challenge. Here, we modulate oxygen coverage on the surface of Ti$_3$C$_2$T$_x$ MXenes to boost the catalytic activity toward HCOOH dehydrogenation. Impressively, Ti$_3$C$_2$T$_x$ MXenes after treating with air at 250 °C (Ti$_3$C$_2$T$_x$-250) significantly increase the amount of surface oxygen atoms without the change of crystalline structure, exhibiting a mass activity of 365 mmol·g$^{-1}$·h$^{-1}$ with 100% of selectivity for H$_2$ at 80 °C, which is 2.2 and 2.0 times that of commercial Pd/C and Pt/C, respectively. Further mechanistic studies demonstrate that HCOO* is the intermediate in HCOOH dehydrogenation over Ti$_3$C$_2$T$_x$ MXenes with different coverages of surface oxygen atoms. Increasing the oxygen coverage on the surface of Ti$_3$C$_2$T$_x$ MXenes not only promotes the conversion from HCOO* to CO$_2$* by lowering the energy barrier, but also weakens the adsorption energy of CO$_2$ and H$_2$, thus accelerating the dehydrogenation of HCOOH.

[1] State Key Laboratory for Powder Metallurgy, Key Laboratory of Electronic Packing and Advanced Functional Materials of Hunan Province, School of Materials Science and Engineering, Central South University, 410083 Changsha, Hunan, P. R. China. [2] Institutes of Physical Science and Information Technology, Anhui University, 230601 Hefei, P.R. China. [3] Hefei National Laboratory for Physical Sciences at the Microscale, University of Science and Technology of China, 230026 Hefei, P. R. China. [4] Key Laboratory of Soil Environment and Pollution Remediation, Institute of Soil Science, the Chinese Academy of Sciences, 210008 Nanjing, Jiangsu, P. R. China. [5] State Key Laboratory of Environment-friendly Energy Materials, Southwest University of Science and Technology, 621010 Mianyang, Sichuan, P. R. China. [6] These authors contributed equally: Tingting Hou, Qiquan Luo, Qi Li. ✉email: linyue@ustc.edu.cn; jlyang@ustc.edu.cn; wanglb@csu.edu.cn

A s an appealing hydrogen carrier in liquid state, formic acid (HCOOH) has attracted tremendous attention due to its great potential for applications in fuel cells and chemical engineering[1–3]. Noble-metal-based catalysts are widely used in the dehydrogenation of HCOOH, exhibiting remarkable activity and selectivity[4–13]. For instance, the turnover frequency (TOF) reached 2882 $h^{-1}$ in pure HCOOH at 50 °C over an Au catalyst modified with Schiff base[7]. The well-dispersed NiPd nanocrystals supported on nitrogen-functionalized graphene displayed superior activity with the TOF of 954.3 $h^{-1}$ for HCOOH decomposition at room temperature[8]. Very recently, our group developed $Pt_1$/Te single-atom catalysts, which delivered the TOF of 1206 $h^{-1}$ with 100% selectivity for $H_2$ at 25 °C via a plasmon-enhanced catalytic process[14]. In spite of these advances, the expensive cost of noble-metal-based catalysts impels people to exploit non-noble-metal-based catalysts for HCOOH decomposition, which has achieved exciting advances recently, especially for homogeneous catalysts[15–24]. For example, Beller et al.[17] reported that a Fe catalyst with an extra equivalent of tetradentate auxiliary ligand decomposed HCOOH at 80 °C with the TOF of 9425 $h^{-1}$. In the presence of pincer ligands, Fe catalysts exhibited ~1,000,000 turnovers assisted by Lewis acid[16]. As for recyclable heterogeneous catalysts, the activity of non-noble-metal-based nanocrystals for HCOOH dehydrogenation was far below than that of their homogeneous counterparts. Therefore, developing non-noble-metal heterogeneous catalysts with high efficiency of HCOOH dehydrogenation is urgently desired but remains a great challenge.

Incorporation of nonmetallic elements, such as C, N, O, and S, into non-noble-metal heterogeneous catalysts has emerged as an effective approach to improve their catalytic performance from both theoretical and experimental perspectives[14,25–31]. The incorporation of nonmetallic elements regulates the electronic structure of catalysts. For example, the introducing of N atoms effectively manipulated the surface electron densities of $NiCo_2S_4$, leading to an outstanding activity comparable to platinum on carbon (Pt/C) in hydrogen evolution[30]. Besides, defective structures generate accompanied with the involvement of nonmetallic elements in nanocrystals. A notable example is the fabrication of $Ti^{3+}$ species and oxygen vacancies in $TiO_2$ nanocrystals by doping N atoms[25–27]. Moreover, the reaction paths are even altered after the introduction of nonmetallic elements. For instance, N atoms in Co-based nanosheets directly interacted with $CO_2$ and $H_2$ to form HCOO* intermediates in $CO_2$ hydrogenation[28]. Hence, modification of non-noble-metal heterogeneous catalysts with nonmetallic atoms serves as a potential route to construct efficient catalysts in HCOOH dehydrogenation.

Herein we develop a highly active non-noble-metal heterogeneous catalyst for HCOOH dehydrogenation, prepared by modulating the oxygen coverage on the surface of $Ti_3C_2T_x$ MXenes. $Ti_3C_2T_x$ MXenes after treating with air at 250 °C ($Ti_3C_2T_x$-250) significantly increase the amount of surface oxygen atoms without the change of crystalline structure, delivering 94% of conversion for HCOOH with 100% of selectivity for $H_2$ at 80 °C. In addition, the mass activity of $Ti_3C_2T_x$-250 is 365 mmol $g^{-1} h^{-1}$, 2.2 and 2.0 times that of commercial palladium on carbon (Pd/C) and Pt/C, respectively. Further mechanistic studies demonstrate that HCOO* is the intermediate in HCOOH dehydrogenation over $Ti_3C_2T_x$ MXenes with different coverages of surface oxygen atoms. Increasing the oxygen coverage on the surface of $Ti_3C_2T_x$ MXenes not only promotes the conversion from HCOO* to $CO_2$* by lowering the energy barrier but also weakens the adsorption energy of $CO_2$ and $H_2$, thus accelerating the dehydrogenation of HCOOH.

## Results

**Synthesis and structural characterizations.** To begin with, $Ti_3C_2T_x$ MXenes ($T_x$ represents the adsorbed species on the surface) were synthesized by immersing commercial $Ti_3AlC_2$ powders in hydrofluoric acid (HF) for 72 h at room temperature, followed by ultrasonic treatment (Supplementary Fig. 1). As shown by the X-ray diffraction (XRD) patterns in Supplementary Fig. 2, the diffraction peaks for $Ti_3AlC_2$ disappeared after etching, with the generation of the peaks for $Ti_3C_2T_x$ MXenes (denoted as $Ti_3C_2T_x$-25 for simplification). To modulate the coverage of oxygen atoms on the surface, we treated $Ti_3C_2T_x$-25 in a muffle furnace at 150, 250, and 350 °C for 1 h to acquire $Ti_3C_2T_x$-150, $Ti_3C_2T_x$-250, and $Ti_3C_2T_x$-350, respectively. As shown in the high-angle annular dark-field scanning transmission electron microscopy (HAADF-STEM) images (Fig. 1a and Supplementary Fig. 3), $Ti_3C_2T_x$-150, $Ti_3C_2T_x$-250, and $Ti_3C_2T_x$-350 exhibited a nanosheet morphology which resembled that of $Ti_3C_2T_x$-25. Further observation in the magnified HAADF-STEM image of $Ti_3C_2T_x$-250 displayed that the lattice spacing was 2.6 Å, corresponding to the (0$\bar{1}$10) crystal plane of $Ti_3C_2T_x$ MXenes (Fig. 1b). STEM energy-dispersive X-ray (EDX) elemental mapping images of an individual $Ti_3C_2T_x$-250 nanosheet indicated the homogeneous distribution of Ti, O, C, F, and Al elements (Fig. 1c). Inductively coupled plasma atomic emission spectroscopy (ICP-AES) result revealed that the mass ratio of Al in $Ti_3C_2T_x$-250 was about 2.0%. In addition, the crystalline structure of $Ti_3C_2T_x$-150 and $Ti_3C_2T_x$-250 was quite similar to that of $Ti_3C_2T_x$-25 based on XRD analysis (Supplementary Fig. 2). As a comparison, the XRD pattern of $Ti_3C_2T_x$-350 was indexed to a mixture of $Ti_3C_2T_x$ MXene phase and tetragonal $TiO_2$ phase (PDF#65-5714). Besides, the Brunauer–Emmett–Teller (BET) surface areas of $Ti_3C_2T_x$-25, $Ti_3C_2T_x$-150, $Ti_3C_2T_x$-250, and $Ti_3C_2T_x$-350 were determined to be 16, 20, 21, and 44 $m^2 g^{-1}$, respectively (Supplementary Table 1). To determine the electronic and local structures of chemical nature of these samples, the X-ray absorption near-edge structure (XANES) spectrum and the extended X-ray absorption fine structure (EXAFS) spectrum were recorded. As shown in Ti K-edge XANES profiles (Fig. 1d), the absorption edge position of $Ti_3C_2T_x$-25, $Ti_3C_2T_x$-150, and $Ti_3C_2T_x$-250 were between those of Ti foil and $TiO_2$. As a result, Ti species in $Ti_3C_2T_x$-25, $Ti_3C_2T_x$-150, and $Ti_3C_2T_x$-250 were partially oxidized. As for $Ti_3C_2T_x$-350, the edge position was similar to that of $TiO_2$, proving that $TiO_2$ phase made up a dominated part in $Ti_3C_2T_x$-350, consistent with the XRD results. This result was further confirmed by wavelet transform (WT) analysis (Fig.1e, f and Supplementary Fig. 4). Figure 1g presented Fourier-transformed EXAFS spectra in R space. The Ti-C coordination and Ti-O coordination were difficult to distinguish apart in EXAFS measurement due to the similar electron scattering abilities of O and C atoms. Thus we calculated Ti-C coordination plus Ti-O coordination (denoted it as Ti-C/O). As shown in Supplementary Table 2, the coordination numbers (CNs) for Ti-C/O shell in $Ti_3C_2T_x$-25, $Ti_3C_2T_x$-150, and $Ti_3C_2T_x$-250 were determined to be 4.5, 4.7, and 4.9, respectively. As for $Ti_3C_2T_x$-350, the CN for Ti-C/O shell was determined to be 2.9. Furthermore, we also performed Al K-edge XANES spectra of $Ti_3C_2T_x$-250 (Supplementary Fig. 5). Obviously, Al species in $Ti_3C_2T_x$-250 exhibited the characteristic of oxidation state.

We further explored the electronic properties of the catalysts by X-ray photoelectron spectroscopy (XPS) and the diffuse reflectance ultraviolet visible (UV-Vis) spectra. As shown in the O 1s spectra (Fig. 2a), four peaks centered at 533.7, 532.2, 531.7, and 529.7 eV were ascribed to the surface-adsorbed oxygen species, surface oxygen atoms bonded to titanium atoms (denoted as surface O-Ti species), surface OH groups, and lattice oxygen in

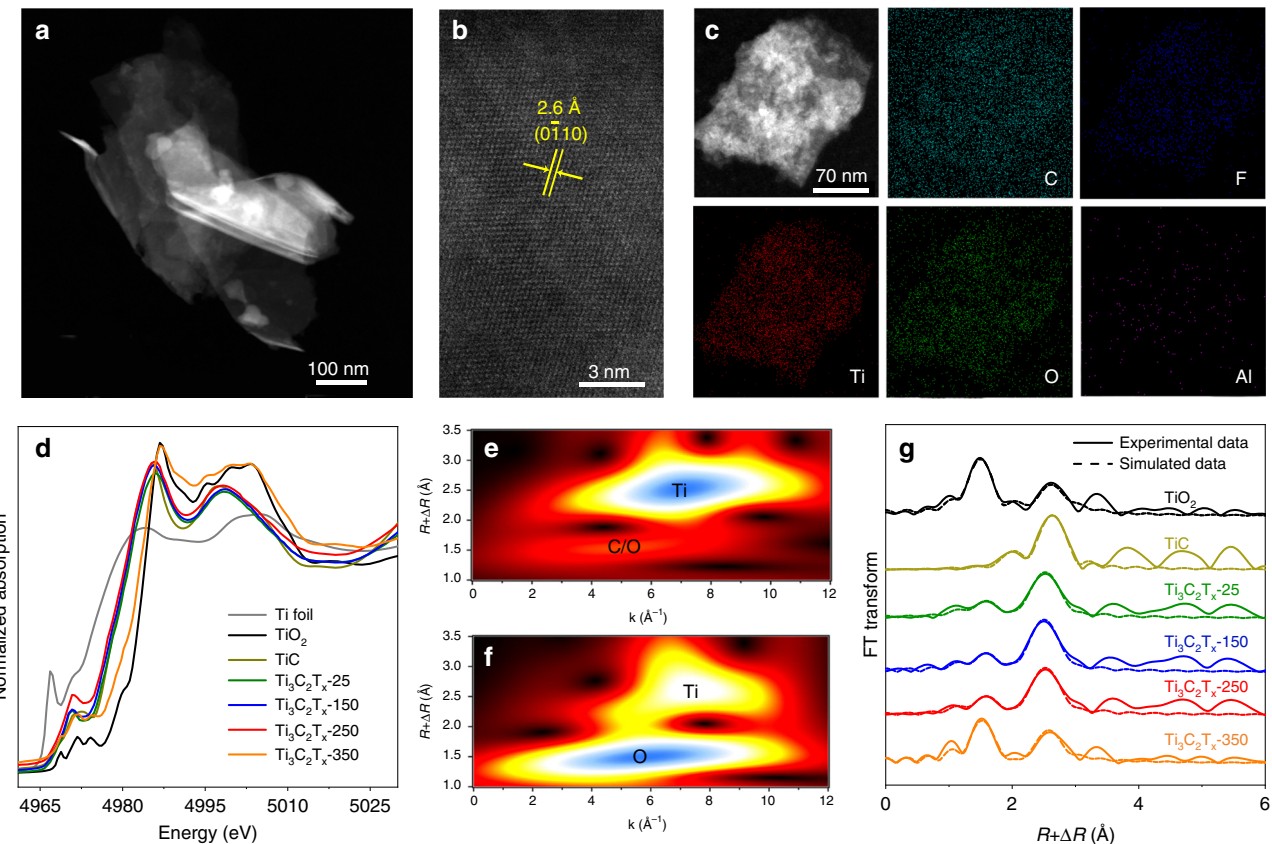

**Fig. 1 Characterization of the obtained samples. a** HAADF-STEM and **b** magnified HAADF-STEM images of $Ti_3C_2T_x$-250. **c** STEM-EDX elemental mapping of $Ti_3C_2T_x$-250. **d** Ti $K$-edge XANES spectra for $Ti_3C_2T_x$-25, $Ti_3C_2T_x$-150, $Ti_3C_2T_x$-250, and $Ti_3C_2T_x$-350. The wavelet transform analysis in Ti $K$-edge for **e** $Ti_3C_2T_x$-250 and **f** $Ti_3C_2T_x$-350. **g** EXAFS spectra in $R$ space for $Ti_3C_2T_x$-25, $Ti_3C_2T_x$-150, $Ti_3C_2T_x$-250, and $Ti_3C_2T_x$-350. Ti foil, $TiO_2$, and TiC were used as references.

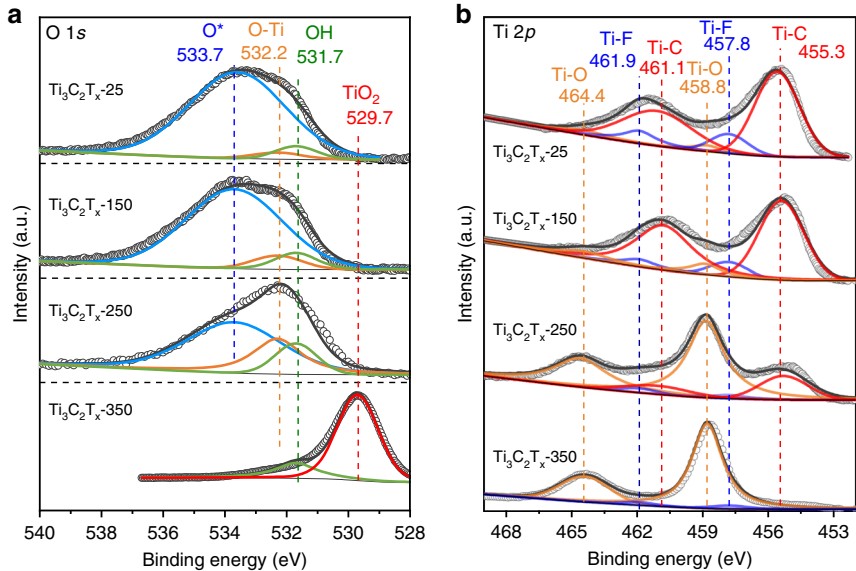

**Fig. 2 XPS spectra of the obtained samples. a** O 1s and **b** Ti 2p XPS spectra for $Ti_3C_2T_x$-25, $Ti_3C_2T_x$-150, $Ti_3C_2T_x$-250, and $Ti_3C_2T_x$-350.

$TiO_2$, respectively[32]. The presence of surface OH groups was further confirmed by diffuse reflectance infrared Fourier transform (DRIFT) measurements (Supplementary Fig. 6). Surface-adsorbed oxygen species was considered as the physical adsorption. Thus surface hydroxyl groups and surface O-Ti species were regarded as the covered oxygen on the surface.

Apparently, the O 1s spectrum of $Ti_3C_2T_x$-150 was quite similar to that of $Ti_3C_2T_x$-25, except for a slight increase in the area of the peak for surface O-Ti species. As for $Ti_3C_2T_x$-250, the peak intensity of surface O-Ti species was significantly strengthened. Besides, a dominated peak for $TiO_2$ appeared for $Ti_3C_2T_x$-350. In addition, we also estimated the relative proportion of O-Ti species

of Ti₃C₂Tₓ-25, Ti₃C₂Tₓ-150, and Ti₃C₂Tₓ-250 according to O 1s XPS spectra. As shown in Supplementary Table 3, after peak deconvolution, the percentages of O-Ti species were estimated to be 3.4, 7.3, and 26.5% for Ti₃C₂Tₓ-25, Ti₃C₂Tₓ-150, and Ti₃C₂Tₓ-250, respectively. Furthermore, Ti 2p XPS spectra were also carried out. As shown in Fig. 2b, the Ti 2p spectra of these samples were fitted with six main constituent peaks, corresponding to Ti-C at 461.1 and 455.3 eV, Ti-O at 464.4 and 458.8 eV, and Ti-F at 461.9 eV and 457.8 eV, respectively[33–37]. As for Ti₃C₂Tₓ-250, the peaks for Ti-O were significantly strengthened compared with those for Ti₃C₂Tₓ-25 and Ti₃C₂Tₓ-150. It is worth noting that the peaks for Ti-F decreased with the elevation of treatment temperatures for Ti₃C₂Tₓ MXenes. With regard to Ti₃C₂Tₓ-350, the peaks for Ti-O became the dominated ones with the neglectable peaks for Ti-C and Ti-F. As a result, the oxygen coverage on the surface of Ti₃C₂Tₓ MXenes could be facilely modulated by thermal treatment below 250 °C without the variation of crystalline structure.

**Catalytic performance of different catalysts.** The catalytic properties of the as-obtained Ti₃C₂Tₓ-25, Ti₃C₂Tₓ-150, Ti₃C₂Tₓ-250, and Ti₃C₂Tₓ-350 were evaluated in comparison with commercial Pt/C and Pd/C toward dehydrogenation of HCOOH (Fig. 3a). Each reaction was performed in a home-made catalytic system containing 2.6 mmol of HCOOH and 10 mL of deionized water at 80 °C. A blank test was conducted without any catalyst, where almost no gas released. When the reaction was then catalyzed by Ti₃C₂Tₓ-25, Ti₃C₂Tₓ-150, and Ti₃C₂Tₓ-350, 16, 33, and 23 mL of gas generated after 40 min, respectively. Under the same reaction condition, the volume of the gas reached 120 mL over Ti₃C₂Tₓ-250. As a reference, commercial Pt/C (5% mass loading) and Pd/C (5% mass loading) were tested, where 60 and 65 mL of gas yielded after 40 min, respectively. Thus Ti₃C₂Tₓ-250 exhibited a significantly enhanced catalytic activity compared to Pt/C and Pd/C catalysts. Typically, HCOOH decomposition undergoes two distinct reaction pathways:

$$HCOOH \rightarrow H_2 + CO_2 \qquad (1)$$

and

$$HCOOH \rightarrow H_2O + CO. \qquad (2)$$

To investigate the selectivity of the catalysts, the generated gases were analyzed by chromatograph (gas chromatography) tests. Only H₂ and CO₂ were detected without the signal of CO (Supplementary Fig. 7). According to Eq. (1), the conversion of HCOOH over Ti₃C₂Tₓ-250 reached 94% within 40 min. Figure 3b illustrates the mass activities for each catalyst within the initial 10 min. The mass activity of Ti₃C₂Tₓ-250 was determined to be 365 mmol g⁻¹ h⁻¹, 7.3, 3.5, 5.7, 2.2, and 2.0 times than that of

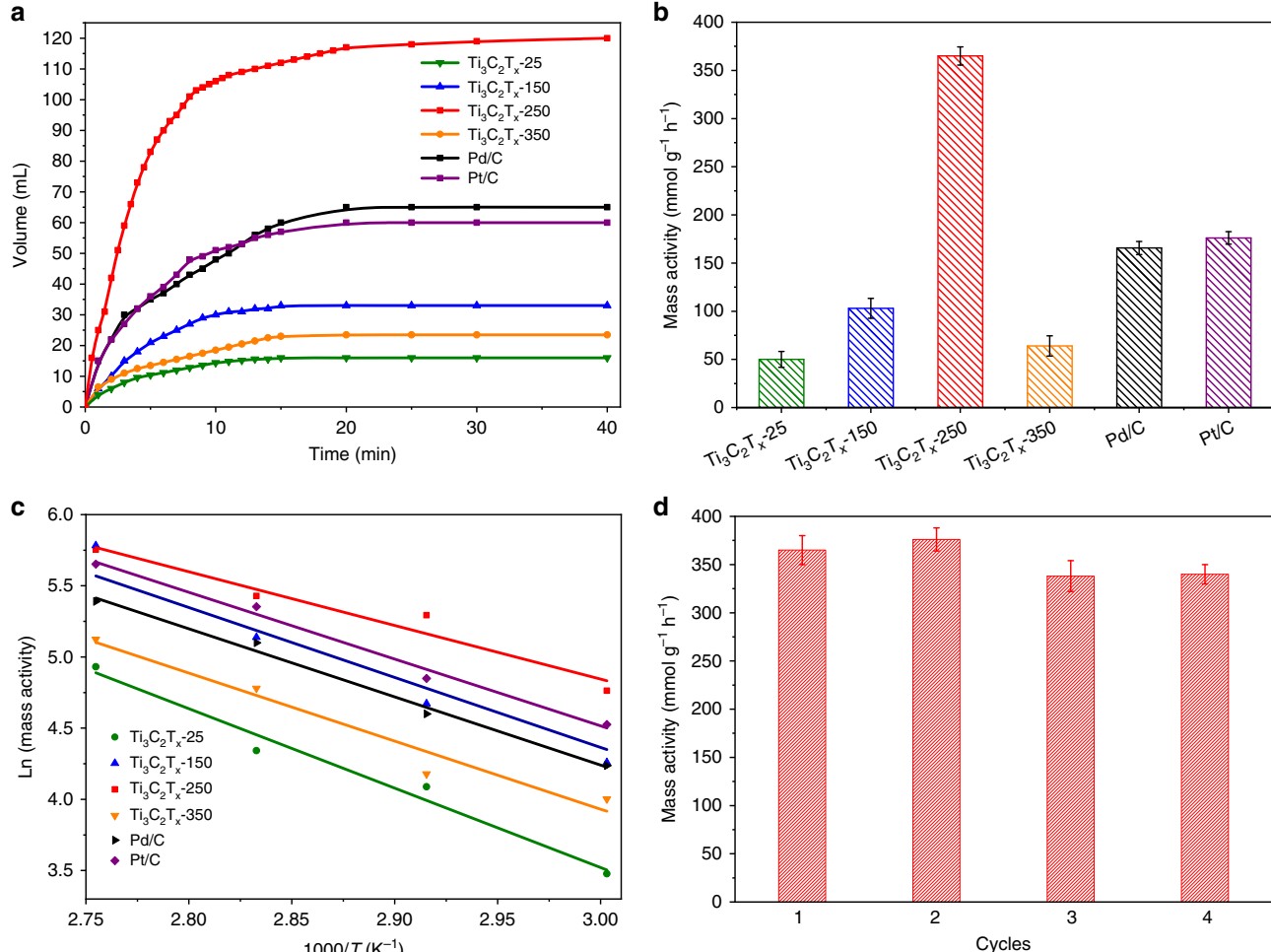

**Fig. 3 Catalytic performance of different catalysts in the dehydrogenation of HCOOH. a** Plots of the volume of gas formed from HCOOH decomposition over different catalysts versus time at 80 °C. **b** Comparison of mass activities for different catalysts. **c** Arrhenius plot for the dehydrogenation of HCOOH over different catalysts. **d** Catalytic performance of Ti₃C₂Tₓ-250 after successive rounds of reaction. The error bar was drawn based on the calculated standard error of three parallel tests.

Ti$_3$C$_2$T$_x$-25, Ti$_3$C$_2$T$_x$-150, Ti$_3$C$_2$T$_x$-350, Pd/C, and Pt/C, respectively. Besides, the turnover frequencies of Ti$_3$C$_2$T$_x$-250, Pd/C, and Pt/C were calculated to be 960, 2679, and 4585 h$^{-1}$, respectively (Supplementary Table 5). Representative works on the dehydrogenation of HCOOH by homogeneous and heterogeneous non-noble-metal catalysts over past few years are listed in Supplementary Table 4[15–17,20,22,24,38–44]. Ti$_3$C$_2$T$_x$-250 was among the best in heterogeneous catalysts. Considering the remaining F$^-$ adsorbed on Ti$_3$C$_2$T$_x$ MXenes, we conducted HCOOH dehydrogenation over Ti$_3$C$_2$T$_x$-250 in NaF aqueous solution with different concentrations to investigate the influence of fluoride species. Despite the varied concentrations of F$^-$, a similar mass activity of ~340 mmol g$^{-1}$ h$^{-1}$ was obtained (Supplementary Fig. 8). To further investigate the effect of the residual Al species on HCOOH dehydrogenation, Ti$_3$C$_2$T$_x$-250-20h and Ti$_3$C$_2$T$_x$-250-96h were prepared with the similar method except for varying the etching time from 72 h to 20 and 96 h, respectively. ICP-AES result revealed that the mass ratios of Al element in Ti$_3$C$_2$T$_x$-250-20h and Ti$_3$C$_2$T$_x$-250-96h were about 3.6 and 1.2%, respectively. When using Ti$_3$C$_2$T$_x$-250-20h and Ti$_3$C$_2$T$_x$-250-96h as the catalysts in HCOOH dehydrogenation, the mass activities were similar to that of Ti$_3$C$_2$T$_x$-250 with the values of 345 and 356 mmol g$^{-1}$ h$^{-1}$ under the same reaction conditions, respectively (Supplementary Fig. 9). Thus the influence of fluoride and the residual Al species on HCOOH dehydrogenation over Ti$_3$C$_2$T$_x$-250 was negligible. Oxygen coverage on the surface of Ti$_3$C$_2$T$_x$ MXenes was regarded as the predominant factor for varied catalytic activity. This point was further confirmed by the relationship between the percentage of O-Ti species and the activity of HCOOH dehydrogenation. As shown in Supplementary Fig. 10a, linear correlation was obtained between the percentage of O-Ti species and the activity of HCOOH dehydrogenation. In addition, the mass activities of Ti$_3$C$_2$T$_x$-250 at different temperatures were further measured to construct the Arrhenius plot (Fig. 3c). The activation energy for the decomposition of HCOOH over Ti$_3$C$_2$T$_x$-250 were

calculated to be 31.4 kJ mol$^{-1}$, which was lower than Ti$_3$C$_2$T$_x$-25, Ti$_3$C$_2$T$_x$-150, Ti$_3$C$_2$T$_x$-350, Pt/C, and Pd/C (Supplementary Table 6)[3,4,6,7,10]. Moreover, the stability of Ti$_3$C$_2$T$_x$-250 was also studied by performing successive rounds of reaction. After four rounds, almost 90% of the original activity was preserved, indicating the high stability of Ti$_3$C$_2$T$_x$-250 (Fig. 3d). This high stability of Ti$_3$C$_2$T$_x$-250 is of crucial importance for potential applications in industrial processes. In addition, we also performed HAADF-STEM, EDX mapping, BET surface areas, and DRIFT measurement of Ti$_3$C$_2$T$_x$-250 after catalytic tests (Ti$_3$C$_2$T$_x$-250-used). As shown in Supplementary Fig. 11, the nanosheet morphology of Ti$_3$C$_2$T$_x$-250-used was well maintained. Besides, the BET surface areas of Ti$_3$C$_2$T$_x$-250 before and after catalytic tests were 21 and 22 m$^2$/g, respectively. In addition, the DRIFT spectrum of Ti$_3$C$_2$T$_x$-250-used still exhibited an obvious absorption peak assigning to the ν(O–H) stretching mode. These results further proved the high stability of Ti$_3$C$_2$T$_x$-250 during catalytic process. Besides, Ti$_3$C$_2$T$_x$-25 was able to be prepared in large-scale, where 6.3305 g of Ti$_3$C$_2$T$_x$-25 was successfully acquired in one pot (Supplementary Fig. 12). Those results provided the possibility for industrial application of Ti$_3$C$_2$T$_x$ MXenes in HCOOH dehydrogenation.

Furthermore, we also investigated the catalytic performance over Ti$_2$CT$_x$ MXenes. Ti$_2$CT$_x$ MXenes were prepared similar to the procedure for Ti$_3$C$_2$T$_x$ MXenes, expect for the use of commercial Ti$_2$AlC powder as the precursor (Supplementary Fig. 13). Ti$_2$CT$_x$-25, Ti$_2$CT$_x$-150, Ti$_2$CT$_x$-250, and Ti$_2$CT$_x$-350 were obtained by thermal treatment at 25, 150, 250, and 350 °C, respectively. Supplementary Fig. 14a showed their catalytic activities in HCOOH dehydrogenation. When the reaction was catalyzed by Ti$_2$CT$_x$-25, Ti$_2$CT$_x$-150, Ti$_2$CT$_x$-250, and Ti$_2$CT$_x$-350, 15, 33, 80, and 42 mL of gas generated after 30 min, respectively. Supplementary Fig. 14b illustrated their mass activities within the initial 10 min. The mass activity of Ti$_2$CT$_x$-250 was determined to be 234 mmol g$^{-1}$ h$^{-1}$, 6.3, 2.3, and 1.8 times as high as that of Ti$_2$CT$_x$-25, Ti$_2$CT$_x$-150, and Ti$_2$CT$_x$-350,

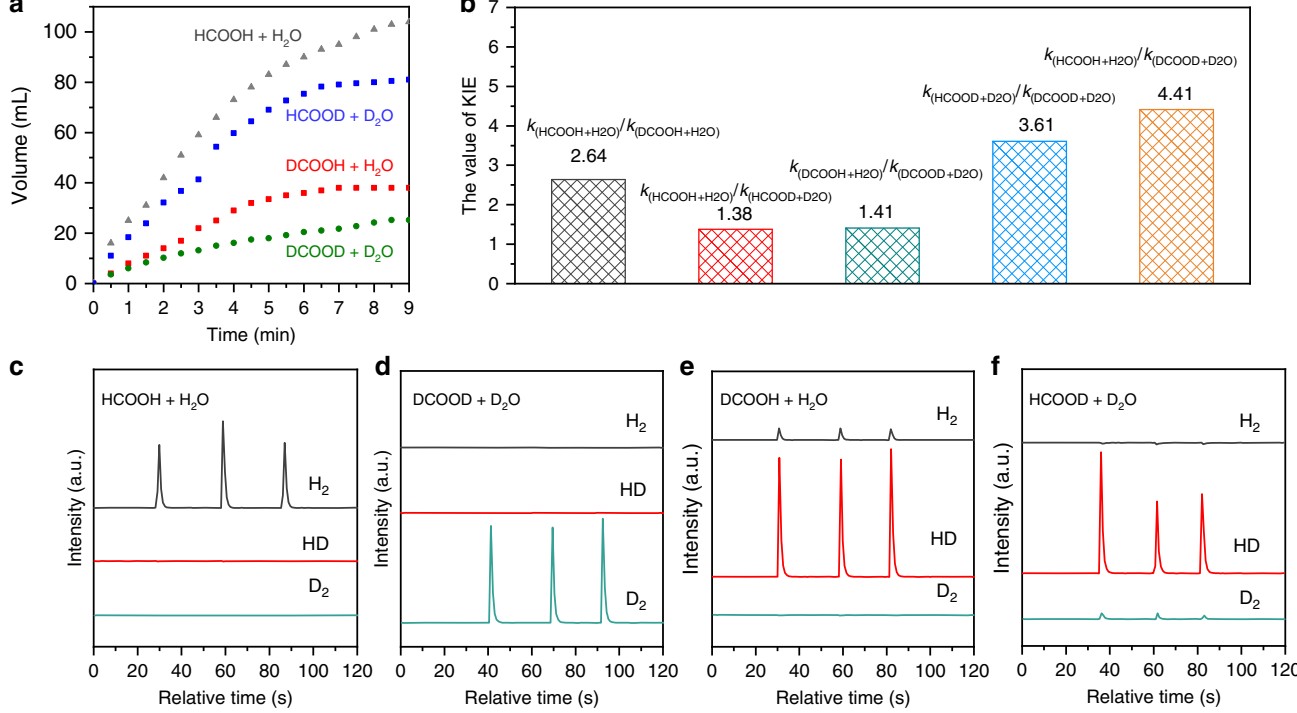

**Fig. 4 Isotope-labeled experiments for dehydrogenation of formic acid over Ti$_3$C$_2$T$_x$-250. a** Time course for the volume of gas generated by employing different isotope-labeled reagents. **b** Kinetic isotope effect data. **c–f** Detection of the gases of H$_2$, HD, and D$_2$ by mass spectrometer.

respectively. The results indicated that the catalytic activity of $Ti_2CT_x$ MXenes in HCOOH dehydrogenation was able to be modulated by varying oxygen coverage on the surface of $Ti_2CT_x$ MXenes. In addition, $Ti_2CT_x$-250 exhibited a comparable catalytic activity with that of $Ti_3C_2T_x$-250 in HCOOH dehydrogenation.

**Investigation of the mechanism for HCOOH dehydrogenation.** To investigate the reaction mechanism in HCOOH dehydrogenation, we investigated the kinetic isotope effect (KIE) of dehydrogenation of HCOOH over $Ti_3C_2T_x$-250. As for HCOOH, the hydrogen in the carboxyl group is able to be exchanged with deuterium oxide at room temperature, while the hydrogen in the methyl group is nearly unable to be exchanged with deuterium oxide under the reaction conditions[45–47]. Thus, to investigate the KIE, we performed the dehydrogenation of HCOOH over $Ti_3C_2T_x$-250 by employing different substrates, including HCOOH, formic-d acid (DCOOH), formic acid-d (HCOOD), and formic acid-d2 (DCOOD, and solvents, including $H_2O$ and $D_2O$. The experiment by using specific HCOOH and water as the substrate and solvent was abbreviated as "formic acid+water." In this case, we implemented the following experiments: HCOOH + $H_2O$, DCOOH + $H_2O$, HCOOD + $D_2O$, and DCOOD + $D_2O$. As the rate of HCOOH dehydrogenation is generally constant at the beginning of the reaction, the initial rate is used in KIE studies for HCOOH dehydrogenation[5]. In this work, the initial rate was named as $k_{(formic\ acid+water)}$ by using specific HCOOH and water as the substrate and solvent, respectively. The reaction profiles versus reaction time are shown in Fig. 4a, where gases were all gradually generated at different reaction conditions. The values of $k_{(HCOOH+H2O)}/k_{(DCOOH+H2O)}$, $k_{(HCOOD+D2O)}/k_{(DCOOD+D2O)}$, $k_{(HCOOH+H2O)}/k_{(HCOOD+D2O)}$, $k_{(DCOOH+H2O)}/k_{(DCOOD+D2O)}$, and $k_{(HCOOH+H2O)}/k_{(DCOOD+D2O)}$ were calculated to be 2.64, 3.61, 1.38, 1.41, and 4.41, respectively (Fig. 4b). Generally speaking, a first-order KIE is obtained when the value of $k_H/k_D$ was greater than ~1.5[48,49]. Thus the first-order KIE was attributed to the dissociation of H/DCOO* to form COO* over $Ti_3C_2T_x$-250, which was regarded as the rate-determining step. Besides, the released gas was also analyzed by mass spectroscopy. As shown in Fig. 4c, d, $H_2$ and $D_2$ were detected as the sole product in the cases of HCOOH + $H_2O$ and DCOOD + $D_2O$, respectively. As for the cases of DCOOH + $H_2O$ and HCOOD + $D_2O$, HD was detected as the dominated product, as well as a small amount of $H_2$ and $D_2$, respectively (Fig. 4e, f). Accordingly, surface hydride and hydrogen atom from HCOO* combined to get $H_2$. A nearly mono-molecular concerted mechanism rather than β-hydride elimination mechanism was regarded to have occurred over $Ti_3C_2T_x$-250 toward the dehydrogenation of HCOOH.

To further investigate the reaction intermediates in HCOOH dehydrogenation, we performed in situ DRIFT measurements. After treatment of the catalysts with a flowing $N_2$ (1 bar) for 20 min at 25 °C, the background spectrum was acquired. Then 1 bar of $N_2$ was allowed to bubble in HCOOH solution and flowed into the cell for 20 min at 25 °C, followed by in situ DRIFT measurements, which was regarded as the in situ DRIFT spectrum at 0 min. Then the cell was sealed and heated to 80 °C within ca. 20 s. In situ DRIFT spectra were then recorded at interval at 80 °C. As for $Ti_3C_2T_x$-250 shown in Fig. 5a, three sets of frequencies were observed once HCOOH was introduced. The set of frequencies at 2945, 1790, 1208, and 1100 $cm^{-1}$ were assigned to the stretching vibration of C-H, the stretching vibration of C=O, the coupled stretching vibration of C-O with the deformation vibration of O-H, and the stretching vibration of C-OH in HCOOH* species, respectively[50]. The set of typical frequencies at 2882, 1555, and 1455 $cm^{-1}$ corresponded to the

stretching vibration of C-H, the asymmetrical stretching vibration of COO, and the symmetrical stretching vibration of COO in HCOO* species, respectively[51–53]. After reaction for 5 min at 80 °C, the peaks for HCOOH* species and HCOO* species significantly attenuated (Fig. 5a). More importantly, the peaks for physisorbed $CO_2^*$ species at 2334 and 2364 $cm^{-1}$ strengthened tremendously. Further prolonging the reaction time to 10 and 20 min, the peaks for HCOOH* species and HCOO* species vanished. Meanwhile, the peaks for physisorbed $CO_2^*$ species became the dominated ones. In comparison, as for $Ti_3C_2T_x$-25, the peaks for HCOOH* species slightly weakened while the peaks for HCOO* species remained almost unchanged (Fig. 5b). In addition, no obvious enhancement in peaks for physisorbed $CO_2^*$ species was observed for $Ti_3C_2T_x$-25 (Fig. 5b). Thus, HCOO* species was regarded as reaction intermediate in HCOOH dehydrogenation. Moreover, it was assumed that HCOOH* could be consumed swiftly while the conversion of HCOO* to $CO_2^*$ was blocked. Therefore, $Ti_3C_2T_x$-250 performed with high efficiency in the transformation from HCOO* to $CO_2^*$, gaining the highest activity in dehydrogenation of HCOOH. By contrast, catalytic reactions over $Ti_3C_2T_x$-25 were obstructed by HCOO*-to-$CO_2^*$ conversion. As for the role of OH groups in catalytic reaction, XPS spectra and DRIFT spectra proved the presence of OH groups on the surface of $Ti_3C_2T_x$ MXenes (Fig. 2a and Supplementary Fig. 6). More importantly, the H atom of HCOOH* was able to dissociate to form HCOO* and OH* on the surface of $Ti_3C_2T_x$ MXenes. Thus the absorption peak ranging from 3000 to 3700 $cm^{-1}$ was obviously observed from in situ DRIFT measurements for both $Ti_3C_2T_x$-25 and $Ti_3C_2T_x$-250 (Fig. 5a, b). As for $Ti_3C_2T_x$-25, the peak of OH* species slightly weakened while the peaks of HCOO* species remained almost unchanged from 0 to 20 min at 80 °C. With regard to $Ti_3C_2T_x$-250, the peak of OH* species vanished from 0 to 20 min at 80 °C with the increasing of the peaks of $CO_2^*$ species, because surface OH* and hydrogen atom from HCOO* combined to get $H_2$ and $CO_2$. Thus the surface OH* groups were nearly unable to promote the dehydrogenation of HCOOH.

To gain atomic-level insight into the origin of remarkable activity for $Ti_3C_2T_x$-250, density functional theory (DFT) calculations were carried out. Based on the experimental characterization results, three models were constructed to mimic different oxygen coverages, i.e., $Ti_3C_2$ without surface oxygen atoms, $Ti_3C_2O_2$-V with oxygen vacancies on the surface, and $Ti_3C_2O_2$ with saturated oxygen coverage (Supplementary Fig. 15), respectively. We screened possible intermediates and optimal paths for HCOOH decomposition over three models. HCOOH adsorbed on $Ti_3C_2$ with an adsorption energy of −3.05 eV (Supplementary Fig. 16), followed by spontaneous dissociation to generate HCOO* and H* with the energy barrier ($E_a$) of only 0.25 eV. Nevertheless, the dissociation of the H atom from HCOO* was impeded thermodynamically with the $E_a$ of 1.65 eV. With regard to $Ti_3C_2O_2$-V, the dissociation of HCOO* species was also inhibited with the $E_a$ of 2.50 eV when the oxygen vacancy served as the active site (Supplementary Fig. 17). As for $Ti_3C_2O_2$, HCOOH exhibited a bidentate adsorption configuration with an adsorption energy of −0.35 eV, followed by the dissociation of H atom to form -OH group (Fig. 5c). Afterwards, HCOO* adjusted the adsorption configuration to form meta-stable state with an increased energy of 0.68 eV. The $E_a$ for the dissociation of HCOO* over $Ti_3C_2O_2$ through meta-stable state was only 0.61 eV, much lower than that over $Ti_3C_2$ and $Ti_3C_2O_2$-V. Due to the presence of the same active sites, HCOOH was speculatively decomposed to $CO_2$ and $H_2$ over $Ti_3C_2O_2$-V similar to the channel over $Ti_3C_2O_2$ when oxygen vacancy was not selected as the reactive site. Besides, we also considered the β-hydride elimination mechanism by calculating the formation of

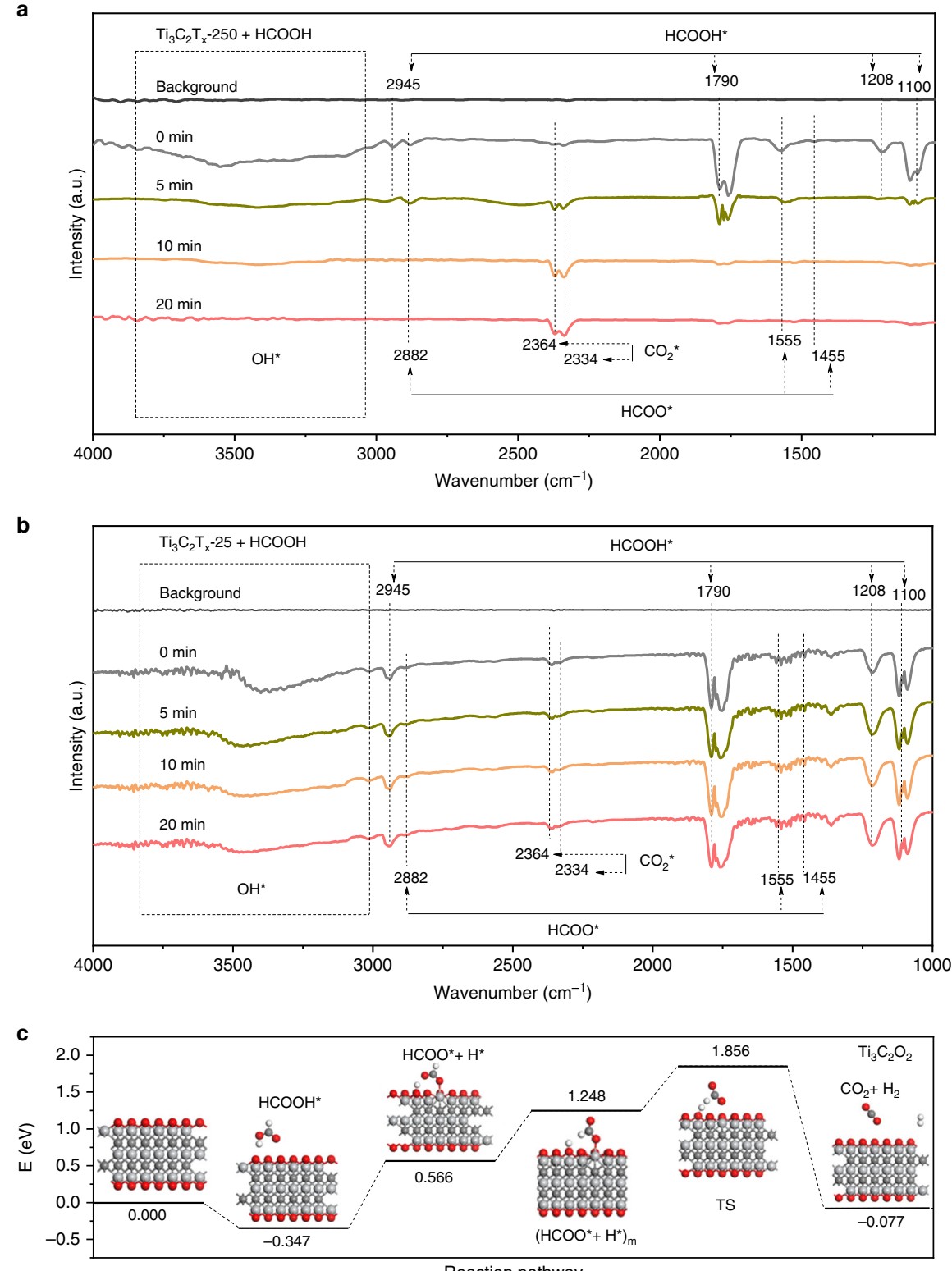

**Fig. 5 Investigation of the reaction mechanism for HCOOH dehydrogenation. a**, **b** In situ DRIFT spectra for $Ti_3C_2T_x$-250 and $Ti_3C_2T_x$-25 after the treatment of HCOOH. **c** Reaction paths for the dehydrogenation of HCOOH over $Ti_3C_2O_2$. Asterisk (*) and subscript m represent the adsorbed species and meta-stable state, respectively.

$H_2$ from two surface H atoms bonded to the surface of $Ti_3C_2O_2$. The result indicated that this process was highly impeded thermodynamically with a reaction energy of 1.10 eV. Based on the above analysis, HCOO* was the intermediate in HCOOH dehydrogenation over all three models with different coverages of

surface oxygen atoms, consistent with the results of in situ DRIFT. Moreover, the energy barrier was lowered with the increasing of coverages of surface oxygen atoms. In addition, the oxygen atoms on the surface of $Ti_3C_2O_2$ also benefited to the desorption of $CO_2$ and $H_2$ with the adsorption energies of −0.16

and −0.06 eV, respectively (Supplementary Table 7). By contrast, the adsorption energies for $CO_2$ and $H_2$ on $Ti_3C_2$ were up to −2.93 and −0.51 eV, respectively (Supplementary Table 7). As a result, the surface oxygen atoms on $Ti_3C_2T_x$ MXenes facilely regulate the reaction energy barriers as well as the adsorption energies of intermediate species, contributing to the enhanced catalytic activity toward HCOOH dehydrogenation. Based on the above analysis, the surface Ti species were regarded as the active sites. Surface oxygen coverage on the surface of $Ti_3C_2T_x$ MXenes regulated the catalytic properties of surface Ti species. Therefore, the possible active species in this case were regarded as surface [O-Ti-C] species.

Besides, considering the presence of other species on the surface of $Ti_3C_2T_x$-250, such as OH, F, and Al, we also investigated their possible effect on HCOOH dehydrogenation from theoretical perspectives. To investigate the effect of surface OH groups, we put a single H atom to pre-adsorb on $Ti_3C_2O_2$ surface ($Ti_3C_2O_2$-$H_1$) as a model (Supplementary Fig. 18a). As shown in Supplementary Fig. 18b, $Ti_3C_2O_2$-$H_1$ exhibited a stronger adsorption for HCOOH with an adsorption energy of −0.53 eV than that of $Ti_3C_2O_2$ with the adsorption energy of −0.35 eV. In addition, we also simulated the process of $H_2$ formation on $Ti_3C_2O_2$-$H_1$, where the activation energy was computed to be 0.57 eV, very close to that over $Ti_3C_2O_2$ with the value of 0.61 eV (Supplementary Fig. 18c). Thus the surface OH groups on $Ti_3C_2O_2$ contributed little to the dehydrogenation of HCOOH.

Furthermore, we also investigated the effect of F species on HCOOH dehydrogenation by DFT calculations. In the simulation, two models were constructed, i.e., single F atoms adsorbed on $Ti_3C_2O_2$ and $Ti_3C_2O_2$-V ($F_1$@$Ti_3C_2O_2$ and $F_1$@$Ti_3C_2O_2$-V) (Supplementary Fig. 19a, b). As shown in Supplementary Fig. 19c, F1@$Ti_3C_2O_2$ exhibited a weak adsorption for HCOOH with the adsorption energy of −0.17 eV. As for $F_1$@$Ti_3C_2O_2$-V, a similar adsorption energy of −0.36 eV was observed compared with that for $Ti_3C_2O_2$-V with the value of −0.35 eV (Supplementary Fig. 19d). Therefore, the influence of F species on HCOOH dehydrogenation was negligible in this case, which was also supported by our experiment of conducting HCOOH dehydrogenation over $Ti_3C_2T_x$-250 in NaF aqueous solution with different concentrations (Supplementary Fig. 8).

The influence of the residual Al species on HCOOH dehydrogenation was also investigated in detail. In the simulation, two models were constructed, i.e., $Al_1$@$Ti_3C_2O_2$ and $Al_1$@$Ti_3C_2O_2$-V, which represent a single Al atom adsorbed on the surface of $Ti_3C_2O_2$ and $Ti_3C_2O_2$-V, respectively (Supplementary Figs. 20a and 21a). As shown in Supplementary Figs. 20b and 21b, HCOOH exhibited a dissociation adsorption on the surface of both $Al_1$@$Ti_3C_2O_2$ and $Al_1$@$Ti_3C_2O_2$-V to generate HCOO-Al* and H* with the adsorption energies of −3.29 and −3.26 eV, respectively. The adsorption of HCOOH was too much strong in the presence of Al element, which was unfavorable for the next reaction. More importantly, the subsequent dissociation of HCOO-Al* to COO-Al* over $Al_1$@$Ti_3C_2O_2$ and $Al_1$@$Ti_3C_2O_2$-V was severely impeded thermodynamically with the reaction energies up to 2.46 and 2.54 eV, respectively (Supplementary Fig. 20c and Supplementary Fig. 21c). The $E_a$ for the dissociation of HCOO* over $Ti_3C_2O_2$ through meta-stable state was only 0.61 eV on the surface of $Ti_3C_2O_2$. With regard to $Ti_3C_2O_2$-V, the $E_a$ of the dissociation of HCOO* was 2.5 eV. Therefore, the theoretical results as well as experimental findings indicated that the promotional role of the residual Al species in $Ti_3C_2T_x$-250 could be neglected for HCOOH dehydrogenation.

## Discussion

In conclusion, we have constructed non-noble-metal heterogeneous catalysts with high efficiency for HCOOH dehydrogenation by modulating the surface oxygen coverage of $Ti_3C_2T_x$ MXenes. $Ti_3C_2T_x$-250 exhibited a mass activity of 365 mmol $g^{-1}$ $h^{-1}$ with 100% of selectivity for $H_2$, much higher than that of commercial Pd/C and Pt/C. The remarkable catalytic performance of $Ti_3C_2T_x$ MXenes originates from the lowered reaction energy barrier for HCOO*-to-$CO_2$* step and adsorption energy for $CO_2$ and $H_2$ by the regulation of surface oxygen atoms. This work not only provides a strategy for developing efficient non-noble-metal catalysts but also advances the understanding in precise control of surface structure of catalyst from atomic insight.

## Methods

**Chemicals and materials**. $Ti_3AlC_2$ powder, HCOOH (98%), and HF (≥40%) were obtained from Sinopharm Chemical Reagent Co., Ltd. Pt/C (5% mass loading) and Pd/C (5% mass loading) were obtained from Sigma-Aldrich. DCOOH (95 wt.% in $H_2O$, 98 atom% D), HCOOD (95 wt.% in $D_2O$, 98 atom% D), and DCOOD (95 wt. % in $D_2O$, 98 atom% D) were purchased from Sigma-Aldrich Co. All solvents and chemicals were of analytical grade and used as received without further purification. All aqueous solutions were prepared using deionized water with a resistivity of 18.2 MΩ $cm^{-1}$.

**Preparation of $Ti_3C_2T_x$-25**. In all, 1.225 g of $Ti_3AlC_2$ powder was immersed in 15 mL of 40% HF solution, followed by stirring for 72 h at room temperature. Al species in $Ti_3AlC_2$ powder were selectively etched in this process. After that, the obtained powder was washed with ethanol and water for several times until the pH of the supernatant was elevated to 6.0. Then the product was collected by centrifugation at ~2739 × g for 10 min and dried at 60 °C under vacuum for 24 h. Finally, the as-obtained powder was dispersed into 50 mL of water and subjected to ultra-sonication for 24 h to gain $Ti_3C_2T_x$-25.

**Preparation of $Ti_3C_2T_x$-25 in large scale**. In all, 8.0 g of $Ti_3AlC_2$ powder was immersed in 120 mL of 40% HF solution, followed by stirring for 72 h at room temperature. Al species in $Ti_3AlC_2$ powder was selectively etched in this process. After that, the obtained powder was washed with ethanol and water for several times until the pH of the supernatant was elevated to 6.0. Then the product was collected by centrifugation at ~2739 × g for 10 min and dried at 60 °C under vacuum for 24 h. Finally, the as-obtained powder was dispersed into 400 mL of water and subjected to ultra-sonication for 24 h to gain 6.3305 g of $Ti_3C_2T_x$-25. This successful large-scale preparation of $Ti_3C_2T_x$-25 provided the possibility for industrial application.

**Preparation of $Ti_3C_2T_x$-150, $Ti_3C_2T_x$-250, and $Ti_3C_2T_x$-350**. The as-obtained $Ti_3C_2T_x$-25 powder was treated in a muffle furnace with air at 150, 250, and 350 °C for 1 h to gain $Ti_3C_2T_x$-150, $Ti_3C_2T_x$-250, and $Ti_3C_2T_x$-350, respectively. $Ti_3C_2T_x$-250-20h and $Ti_3C_2T_x$-250-96h were prepared with the similar method except for varying the etching time from 72 h to 20 and 96 h, respectively.

**Preparation of $Ti_2CT_x$-25**. In all, 5.0 g of $Ti_2AlC$ powder was immersed in 30 mL of 16% HF solution, followed by stirring for 72 h at room temperature. Al species in $Ti_2AlC$ powder was selectively etched in this process. After that, the obtained powder was washed with ethanol and water for several times until the pH of the supernatant was elevated to 6.0. Then the product was collected by centrifugation at ~2739 × g for 10 min and dried at 60 °C under vacuum for 24 h. Finally, the as-obtained powder was dispersed into 50 mL of water and subjected to ultra-sonication for 24 h to gain $Ti_2CT_x$-25.

**Preparation of $Ti_2CT_x$-150, $Ti_2CT_x$-250, and $Ti_2CT_x$-350**. The as-obtained $Ti_2CT_x$-25 powder was treated in a muffle furnace with air at 150, 250, and 350 °C for 1 h to gain $Ti_2CT_x$-150, $Ti_2CT_x$-250, and $Ti_2CT_x$-350, respectively.

**X-ray absorption fine structure (XAFS) measurements for Ti element**. The XAFS spectra at Ti K-edge ($E_0$ = 4966 eV) were performed at BL14W1 beamline of Shanghai Synchrotron Radiation Facility (SSRF) operated at 3.5 GeV under "top-up" mode with a constant current of 220 mA. The XAFS data of $Ti_3C_2T_x$-25, $Ti_3C_2T_x$-150, $Ti_3C_2T_x$-250, and $Ti_3C_2T_x$-350 samples were recorded under transmission mode with two ion chambers. The white light was monochromatized by a Si (111) double-crystal monochromator and calibrated with a Ti foil Athena and Artemis codes were used to extract the data and fit the profiles. For the XANES part, the experimental absorption coefficients as a function of energies μ(E) were processed by background subtraction and normalization procedures. We refer to

this process as "normalized absorption." For the EXAFS part, the Fourier-transformed (FT) data were fitted in $R$ space. The passive electron factors, $S_0^2$, were determined by fitting the experimental $TiO_2$ data and fixing the Ti-O CN to be 6 and then fixed for further analysis of the measured samples. The parameters describing the local structure environment including CN, bond distance ($R$), and Debye–Waller factor around the absorbed atoms were allowed to vary during the fit process. WT analysis was employed using the Igor pro script developed by Funke et al.[54]. The Morlet wavelet was chosen as basis mother wavelet and the parameters ($\eta = 8$, $\sigma = 1$) were used for a better resolution in the wave vector $k$.

**XPS measurements**. XPS experiments were performed at the Photoemission Endstation connected to the BL10B beamline in the National Synchrotron Radiation Laboratory in Hefei, China. The beamline is connected to an in-vacuum undulator and equipped with two gratings that offer soft X-rays from 100 to 1000 eV with a typical photon flux of $1 \times 10^{10}$ photons s$^{-1}$. The analysis chamber is connected to the beamline and equipped with a VG Scienta R3000 electron energy analyzer, a twin anode X-ray source (Mg $K\alpha$ and Al $K\alpha$), a UV light source, a rear-view optics for low-energy electron diffraction, and a high-precision manipulator with four degree of freedom. All of the XPS data were fitted by "XPS Peak Fitting Program for WIN95/98 XPSPEAK Version 4.1" developed by Raymund W.M. Kwok. During the process of fitting, the peak is determined by the parameters of peak position, area, full width at half maximum (FWHM), and %Gaussian–Lorentzian. For the fitting peaks, the values of FWHM for one species in different samples should be kept the same, at least <10% (user manual written by Raymund W.M. Kwok, July 1, 1999).

**Catalytic tests**. Catalytic reactions were carried out in a 100-mL three-neck flask immersed into an oil bath. The volume of the gas generated during the catalytic reaction was monitored by a gas burette system. In a typical reaction, 30 mg of catalysts were added into a 100-mL three-neck flask containing 2.6 mmol of HCOOH and 10 mL of deionized water, followed by magnetic stirring at 80 °C for 40 min. The composition of released gas was determined by gas chromatograph (Shimadzu GC-2014C). The mass activity of each catalyst was calculated within the initial 10 min. Besides, after finishing the reaction, the catalyst was recollected by centrifugation for 10 min and dried at 60 °C under vacuum for 12 h, which was named as $Ti_3C_2T_x$-250-used.

**Isotope-labeled experiments**. To acquire KIE data, we performed the dehydrogenation of HCOOH over $Ti_3C_2T_x$-250 under the same conditions excepting for employing different substrates, including HCOOH, DCOOH, HCOOD, and DCOOD, and solvents, including $H_2O$ and $D_2O$. The experimental process is described in detail. (1) 30 mg of $Ti_3C_2T_x$-250 were added into a 100-mL three-neck flask containing 2.6 mmol of HCOOH and 10 mL of deionized water ($H_2O$), followed by magnetic stirring at 80 °C for 10 min. The volume of the released gas was determined by a gas burette system. (2) 30 mg of $Ti_3C_2T_x$-250 were added into a 100-mL three-neck flask containing 2.6 mmol of DCOOH and 10 mL of deionized water ($H_2O$), followed by magnetic stirring at 80 °C for 10 min. The volume of the released gas was determined by a gas burette system. (3) 30 mg of $Ti_3C_2T_x$-250 were added into a 100-mL three-neck flask containing 2.6 mmol of HCOOD and 10 mL of $D_2O$, followed by magnetic stirring at 80 °C for 10 min. The volume of the released gas was determined by a gas burette system. (4) 30 mg of $Ti_3C_2T_x$-250 were added into a 100-mL three-neck flask containing 2.6 mmol of DCOOD and 10 mL of $D_2O$, followed by magnetic stirring at 80 °C for 10 min. The volume of the released gas was determined by a gas burette system. To analyze the generation of $H_2$, HD, and $D_2$, we performed the experiments under the same reaction conditions except for using a 100-mL home-made reactor with gas inlet. The formed $H_2$, $D_2$, and HD were analyzed by mass spectrometry (THERMO$^{Star}$ gas analysis system).

**In situ DRIFT measurements**. In situ DRIFT experiments were conducted in an elevated-pressure cell (Harrick DRK-4-BR4) with a FT infrared spectrometer (Bruker TENSOR II). After treatment of the catalysts with a flowing of $N_2$ (1 bar) for 20 min at 25 °C, the background spectrum was acquired. Then 1 bar of $N_2$ was allowed to bubble in HCOOH solution and flowed into the cell for 20 min at 25 °C, followed by in situ DRIFT measurements, which was regarded as the in situ DRIFT spectrum at 0 min. Then the cell was sealed and heated to 80 °C within ca. 20 s. In situ DRIFT spectra were then recorded at interval at 80 °C.

**DFT calculations**. The calculations were done using the periodic DFT method in the Vienna ab initio simulation package[55]. The electron ion interaction was treated with the projector augmented wave method[56]. The electron exchange and correlation energy was described using the generalized gradient approximation-based Perdew–Burke–Erzenhorf functional[57]. The van der Waals correction was considered within the empirical Grimme scheme (vdW-D3)[58]. The cut-off energy was set up 500 eV. The force convergence was set to be <0.02 eV Å$^{-1}$, and the total energy convergence was set to be <$10^{-5}$ eV. Electron smearing of $\sigma = 0.1$ eV was used following the Gaussian scheme. Brillouin zone sampling was employed using a Monkhorst–Pack grid[59]. The nudged elastic band method was used to locate the transition state between the initial and final states[60].

A single-layer periodic slab with ($3 \times 3$) supercell $Ti_3C_2$ and $Ti_3C_2O_2$ and $Ti_3C_2O_2$-V with exposed (111) facet was employed to model the experimental catalyst, and all the atoms were fully relaxed during the calculations. The vacuum slab was set up to 15 Å. It is noted that $Ti_3C_2$ and $Ti_3C_2O_2$ represent the catalysts without surface oxygen atoms and surface with saturated oxygen coverage, respectively. $Ti_3C_2O_2$-V is the model that has one O atom missing at each surface of the model $Ti_3C_2O_2$, representing a catalyst with less oxygen coverage compared to $Ti_3C_2O_2$.

**Instrumentations**. TEM, HAADF-STEM, and STEM-EDX images were collected on a JEOL ARM-200F field-emission transmission electron microscope operating at 200 KV accelerating voltage. XRD pattern was recorded by using a Philips X'Pert Pro Super diffractometer with Cu-K$\alpha$ radiation ($\lambda = 1.54178$ Å). ICP-AES (Atomscan Advantage, Thermo Jarrell Ash, USA) was used to determine the concentration of Al. The BET surface areas of the samples were measured on a Micromeritics ASAP 2460 adsorption apparatus. The numbers of surface Pd and Pt in Pd/C and Pt/C were determined by CO pulse chemisorption (VDsorb-91i). UV-Vis tests were conducted on a TU-1901 at room temperature. The Al $K$-edge XANES was performed at 08U1A beamline of SSRF.

## Data availability

All data generated and analyzed during this study are included in this article and its Supplementary Information or are available from the corresponding authors upon reasonable request.

## Code availability

All the codes used in the simulations supporting this article are available from the corresponding authors upon request.

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

## Acknowledgements

This work was supported by the National Natural Science Foundation of China (Grant nos. 51801235, 11875258, 11505187, 51374255, 51802356, 51572299, 21688102, 21703222, and 11874334), Innovation-Driven Project of Central South University (No. 2018CX004), the Start-up Funding of Central South University (No. 502045005), the Fundamental Research Funds for the Central South Universities (Nos. WK2310000066, WK2060190081, and WK2060190103), China Postdoctoral Science Foundation (No. 2019M652797), Central South University Postdoctoral Research Opening Fund, Youth Innovation Promotion Association CAS (2020458), joint funding supports from National Synchrotron Radiation Laboratory (KY2340000115), National Key Research, Development Program of China (2019YFA0307900), and Natural Science Foundation of Hunan Province (2020JJ5690). We acknowledge the staff of 08U1A at Shanghai Synchrotron Radiation Facility. The calculations were performed on the supercomputing system at the Supercomputing Center of University of Science and Technology of China and the High-performance Computing Platform of Anhui University.

## Author contributions

T.H. and L.W. designed the studies and wrote the paper. T.H. and Q. Li synthesized catalysts and performed catalytic tests and in situ DRIFT experiment. Q. Luo, J.C., and J.Y. performed DFT calculations. Y.L. performed the aberration-corrected STEM characterization. P.C. and X.Z. conducted XPS and XAFS measurement. H.Z. and S.C. performed XRD experiment. W.Z. and S.L. revised the manuscript. All authors discussed the results and commented on the manuscript.

## Competing interests

The authors declare no competing interests.
