## [Peer Review File · Nature Communications]

Reviewers' comments:

Reviewer #1 (Remarks to the Author):

It is happy to read this nice work which combine synthesis, characterization and evaluation and DFT rationalization. It is nice but not novel, since such type of studies are plentiful in literature both homogeneous and heterogeneous catalysis from experiment and theory. To improve the quality or novelty, additional experiment and computations should be done

The major point is the discussion about the effect of formate, the review would like to see the kinetic isotopic effect (KIE); and the authors can use DCOOH as substrate. This indeed can be used to check the computed mechanism shown in Figure 4.

In Figure 4, the authors present a nearly mono-molecular concerted mechanism, in which surface hydride and H from HCOO combine to get CO₂ and H₂, and this is quite unusual. Normally, HCOO will gives H back to surface via so-called β -hydride elimination, and this is different from the proposed mechanism in current work. If this is true, it would be very interesting

Using HCOO and DCOO one can not only study the KIE and also proof the proposed mechanism (H₂, HD and D₂ formation). There are papers reporting KIE to study the mechanisms.

The authors have to rationalize their use of atomic element (H, Al F) to study the effect of promotion, and this is indeed something non-chemical case. What is the effect if counter ion?

Reviewer #2 (Remarks to the Author):

In this work, the authors modulated oxygen coverage on the surface of Ti₃C₂T_x MXenes to boost the catalytic activity toward HCOOH dehydrogenation. Ti₃C₂T_x MXenes after treating with air at 250 °C (Ti₃C₂T_x-250) significantly increased the amount of surface oxygen atoms, exhibiting high performance for HCOOH dehydrogenation. This work has been done well and the results are sound and important. It can be accepted after minor revision.

- (1) The Toc graphic contains unintentional lines.
- (2) May give more discussions on the effect of Al on the catalyts surface on the catalysis.
- (3) May give clearer HAADF-STEM images for the claimed nanosheet morphology of the Ti₃C₂T_x materials (Fig.1a).

Reviewer #3 (Remarks to the Author):

The presented manuscript is related to utilization of MXenes for the hydrogen production in formic acid decomposition. The authors propose novel catalysts for this reaction and perform their careful characterization. However, there are some serious problems, which do not allow me to advise this paper for publication in Nature Communications. All of them are related to their studies of the reaction mechanism.

They measured BET surface area only for one, the most active sample. Therefore, there is a question, whether their activity difference for the samples can be related to the difference of the surface area. They say that O-Ti concentration increases with the temperature of the treatment in air and this is a reason of the activity increase of their material. However, if to compare the materials treated at 25 and 150 oC. I see a reversed picture. The activity is doubled, but the XPS O-Ti line intensity, in contrast, decreases by a factor of 2. Hence, I cannot relate the activity with these species as they do.

They use IR spectroscopy and say that they observe the bands of formate species at 2882, 1552, 1455 cm⁻¹ in 6 spectra for 3 samples. I have experience in IR spectroscopy and do not see reliably that they have these 3 lines in their spectra. Additional confirmation could be a study of the OH groups region. Some hydroxyls could interact with formic acid and this could be observed in IR spectra. However, the authors do not show this region for the in situ studies.

Also, if they calculate E_a of some steps by DFT can they compare with experimental values of the apparent E_a .

Hence, they need to prove further the presence of formate as this study is a key point for the establishment of the mechanism. They also have to provide more evidence for the dependence of the activity on the specific oxygen species.

Point-by-point response to reviewer comments

Manuscript ID: NCOMMS-19-38445

Reviewer #1 (Remarks to the Author):

“It is happy to read this nice work which combine synthesis, characterization and evaluation and DFT rationalization. It is nice but not novel, since such type of studies are plentiful in literature both homogeneous and heterogeneous catalysis from experiment and theory. To improve the quality or novelty, additional experiment and computations should be done.

1). The major point is the discussion about the effect of formate, the review would like to see the kinetic isotopic effect (KIE); and the authors can use DCOOH as substrate. This indeed can be used to check the computed mechanism shown in Figure 4.”

Thanks for the reviewer’s suggestion. As for formic acid, the hydrogen in the carboxyl group is able to be exchanged with deuterium oxide at room temperature, while the hydrogen in the methyl group is nearly unable to be exchanged with deuterium oxide under the reaction conditions (*J. Chem. Phys.* **1957**, 27 (6), 1305-1308; *The Course of Formic Acid Reduction of Enamines* **1957**, 6210-6214; *J. Chem. Soc.* **1952**, 2125-2127). Thus, to investigate the kinetic isotopic effect (KIE), we have performed the dehydrogenation of formic acid over $Ti_3C_2T_x-250$ by employing different substrates including HCOOH, DCOOH, HCOOD, and DCOOD, and solvents including H_2O and D_2O . The experiment by using specific formic acid and water as the substrate and solvent was abbreviated as “formic acid+water”. In this case, we have implemented the following experiments: HCOOH+ H_2O , DCOOH+ H_2O , HCOOD+ D_2O , and DCOOD+ D_2O . As the rate of formic acid dehydrogenation is generally constant at the beginning of the reaction, the initial rate is used in KIE studies for formic acid dehydrogenation (*Nat. Commun.* **2016**, 7, 11308). In the revised manuscript, the initial rate was named as $k_{(formic\ acid+water)}$ by using specific formic acid and water as the substrate and solvent, respectively. The reaction profiles *versus* reaction time were shown in Fig. 4a, where gases were all gradually generated at different reaction conditions. The values of $k_{(HCOOH+H_2O)}/k_{(DCOOH+H_2O)}$, $k_{(HCOOD+D_2O)}/k_{(DCOOD+D_2O)}$, $k_{(HCOOH+H_2O)}/k_{(HCOOD+D_2O)}$, $k_{(DCOOH+H_2O)}/k_{(DCOOD+D_2O)}$, and $k_{(HCOOH+H_2O)}/k_{(DCOOD+D_2O)}$ were calculated to be 2.64, 3.61, 1.38, 1.41, and 4.41, respectively (Fig. 4b). Generally speaking, a first-order KIE is obtained when the value of k_H/k_D was greater than ~ 1.5 (*ACS Catal.* **2017**, 7, 3850-3859; *Angew. Chem., Int. Ed.* **2012**, 51, 3066-3072). Thus, the first-order KIE was attributed to the dissociation of H/DCOO* to form COO* over $Ti_3C_2T_x-250$, which was regarded as the rate-determining step. The result of KIE studies was well consistent with the catalytic mechanism provided by DFT studies (Figure 5c). We have added some sentences, two figures and six references in the revised manuscript (p. 8, lines 28-31; p. 9, lines 1-17; p. 13, lines 22-24; p. 16, lines 6-22; p. 21, lines 27-31; p. 22, lines 1-8; Figs 4a-4b; highlighted in yellow color).

“2). In Figure 4, the authors present a nearly mono-molecular concerted mechanism, in which surface hydride and H from HCOO combine to get CO_2 and H_2 , and this is quite unusual.

Normally, HCOO will give H back to surface via so-called β -hydride elimination, and this is different from the proposed mechanism in current work. If this is true, it would be very interesting.”

The reviewer raised a very good point. We have considered the β -hydride elimination mechanism by calculating the formation of H₂ from two surface H atoms bonded to the surface of Ti₃C₂O₂. The result indicated that this process was thermodynamically impeded with a reaction energy of 1.10 eV, which was much higher than that of the mechanism proposed in this work with a reaction energy of 0.61 eV. As a result, the proposed mechanism in this work might be more favorable in the case of Ti₃C₂T_x-250. We have added some sentence in the revised manuscript (p. 11, lines 18-21; highlighted in yellow color).

“3). Using HCOO and DCOO one can not only study the KIE and also proof the proposed mechanism (H₂, HD and D₂ formation). There are papers reporting KIE to study the mechanisms.”

We greatly thank this reviewer for his/her suggestions. We have performed the dehydrogenation of formic acid over Ti₃C₂T_x-250 including HCOOH+H₂O, DCOOH+H₂O, HCOOD+D₂O, and DCOOD+D₂O. Besides, the released gas has also been analyzed by mass spectroscopy. As shown in Fig. 4c and Fig. 4d, H₂ and D₂ were detected as the sole product in the cases of HCOOH+H₂O and DCOOD+D₂O, respectively. As for the cases of DCOOH+H₂O and HCOOD+D₂O, HD was detected as the dominated product, as well as a small amount of H₂ and D₂, respectively (Figs. 4e-f). Accordingly, surface hydride and hydrogen atom from HCOO* combined to get H₂. A nearly mono-molecular concerted mechanism rather than β -hydride elimination mechanism was regarded to be occurred over Ti₃C₂T_x-250 towards the dehydrogenation of formic acid. The result of detection of released gas was well consistent with the catalytic mechanism provided by DFT studies (Figure 5c). We have added some sentences as well as four figures in the revised manuscript (9, lines 17-24; p. 13, lines 22-24; p. 16, lines 6-22; Figs 4c-4f; highlighted in yellow color).

“4). The authors have to rationalize their use of atomic element (H, Al F) to study the effect of promotion, and this is indeed something non-chemical case. What is the effect if counter ion”

Thanks for the reviewer’s suggestion. Considering that H, Al and F atoms might exist and adsorb on the surface of catalysts, we computed the catalytic effect of these species correspondingly. Actually, we respectively employed single H, Al and F atoms to adsorb on the surface of models in DFT calculations. Thus, the whole models were always charge-neutral in spite of the charge transfer between these adsorbed atoms and catalysts. In addition, we also tested the charged models in DFT calculations. As we used periodic slab models in calculations, the subtracted or added charges in the models could not be localized at these H, Al or F species, which caused serious convergence problem during the structure optimization. We have added some sentences to

further clarify this point (p. 17, lines 17-21).

Reviewer #2 (Remarks to the Author):

“In this work, the authors modulated oxygen coverage on the surface of $Ti_3C_2T_x$ MXenes to boost the catalytic activity toward HCOOH dehydrogenation. $Ti_3C_2T_x$ MXenes after treating with air at 250 °C ($Ti_3C_2T_x$ -250) significantly increased the amount of surface oxygen atoms, exhibiting high performance for HCOOH dehydrogenation. This work has been done well and the results are sound and important. It can be accepted after minor revision.

1). The Toc graphic contains unintentional lines.”

Thanks for the reviewer’s suggestion. We have removed the unintentional lines from the TOC graphic.

“2). May give more discussions on the effect of Al on the catalyts surface on the catalysis.”

We greatly thank this reviewer for his/her suggestions. In the simulation, two models were constructed, i.e. $Al_1@Ti_3C_2O_2$ and $Al_1@Ti_3C_2O_2-V$, which represented a single Al atom adsorbed on the surface of $Ti_3C_2O_2$ and $Ti_3C_2O_2-V$, respectively (Supplementary Fig. 20a and Supplementary Fig. 21a). As shown in Supplementary Figure 20b and Supplementary Fig. 21b, HCOOH exhibited a dissociation adsorption on the surface of both $Al_1@Ti_3C_2O_2$ and $Al_1@Ti_3C_2O_2-V$ to generate HCOO-Al* and H* with the adsorption energies of -3.29 and -3.26 eV, respectively. The adsorption of HCOOH was too much strong in the presence of Al element, which was unfavourable for the next reaction. More importantly, the subsequent dissociation of HCOO-Al* to COO-Al* over $Al_1@Ti_3C_2O_2$ and $Al_1@Ti_3C_2O_2-V$ was severely impeded thermodynamically with the reaction energies up to 2.46 and 2.54 eV, respectively (Supplementary Fig. 20c and Supplementary Fig. 21c). The E_a for the dissociation of HCOO* over $Ti_3C_2O_2$ through meta-stable state was only 0.61 eV on the surface of $Ti_3C_2O_2$. With regard to $Ti_3C_2O_2-V$, the E_a of the dissociation of HCOO* was 2.5 eV. Therefore, the theoretical results as well as experimental findings indicated that the promotional role of the residual Al species in $Ti_3C_2T_x$ -250 could be neglected for HCOOH dehydrogenation. We have given more discussions on the effect of Al on the catalyts surface on the catalysis and highlighted in yellow in main text (p. 12, lines 26-31; p. 13, lines 1-4).

“3). May give clearer HAADF-STEM images for the claimed nanosheet morphology of the $Ti_3C_2T_x$ materials (Fig. 1a).”

As suggested, we have re-conducted HAADF-STEM measurement for $Ti_3C_2T_x$ -250. A clear HAADF-STEM image has been further provided to claim the formation of nanosheet morphology of $Ti_3C_2T_x$ -250.

Reviewer #3 (Remarks to the Author):

“The presented manuscript is related to utilization of MXenes for the hydrogen production in formic acid decomposition. The authors propose novel catalysts for this reaction and perform their careful characterization. However, there are some serious problems, which do not allow me to advise this paper for publication in Nature Communications. All of them are related to their studies of the reaction mechanism.

1). They measured BET surface area only for one, the most active sample. Therefore, there is a question, whether their activity difference for the samples can be related to the difference of the surface area.”

Thanks for the reviewer’s suggestion. We have performed the Brunauer–Emmett–Teller (BET) surface areas for all samples. As shown in Supplementary Table 1, the surface areas of $\text{Ti}_3\text{C}_2\text{T}_x\text{-25}$, $\text{Ti}_3\text{C}_2\text{T}_x\text{-150}$, $\text{Ti}_3\text{C}_2\text{T}_x\text{-250}$, and $\text{Ti}_3\text{C}_2\text{T}_x\text{-350}$ were determined to be 16, 20, 21, and 44 m^2/g , respectively. The mass activities of $\text{Ti}_3\text{C}_2\text{T}_x\text{-25}$, $\text{Ti}_3\text{C}_2\text{T}_x\text{-150}$, $\text{Ti}_3\text{C}_2\text{T}_x\text{-250}$, and $\text{Ti}_3\text{C}_2\text{T}_x\text{-350}$ were 50, 104, 365, and 64 $\text{mmol}\cdot\text{g}^{-1}\cdot\text{h}^{-1}$, respectively. Thus, their activities were not related to their difference of the surface areas. We have added some sentences as well as one table in the revised manuscript and supplementary information (p. 5, lines 2-5; Supplementary Table 1; highlighted in yellow color).

“2). They say that O-Ti concentration increases with the temperature of the treatment in air and this is a reason of the activity increase of their material. However, if to compare the materials treated at 25 and 150 °C. I see a reversed picture. The activity is doubled, but the XPS O-Ti line intensity, in contrast, decreases by a factor of 2. Hence, I cannot relate the activity with these species as they do.”

We sincerely sorry for the mistake of XPS fitting. In the original manuscript, all of the XPS data was fitted by “XPS Peak Fitting Program for WIN95/98 XPSPEAK Version 4.1” developed by Raymund W.M. Kwok. During the process of fitting, the peak is determined by the parameters of peak position, area, full width at half maximum (FWHM), and %Gaussian-Lorentzian. For the fitting peaks, the values of FWHM for one species in different samples should be kept the same, at least less than 10% (User manual written by Raymund W.M. Kwok, July 1, 1999). Actually, in the original manuscript, the values of FWHM of O-Ti species differed with the difference of 31% for $\text{Ti}_3\text{C}_2\text{T}_x\text{-25}$, $\text{Ti}_3\text{C}_2\text{T}_x\text{-150}$, and $\text{Ti}_3\text{C}_2\text{T}_x\text{-250}$ in O 1s XPS spectra. Thus, we have re-fitted the XPS data after considering all of the parameters. For different samples, the value of FWHM for all peaks in O 1s XPS spectra was kept the same in the revised manuscript. Besides, we also estimated the relative proportion of O-Ti species of $\text{Ti}_3\text{C}_2\text{T}_x\text{-25}$, $\text{Ti}_3\text{C}_2\text{T}_x\text{-150}$ and $\text{Ti}_3\text{C}_2\text{T}_x\text{-250}$ according to O 1s XPS spectra. As shown in Figure 2a and Supplementary Table 3, after peak deconvolution, the percentages of O-Ti species were estimated to be 3.4%, 7.3%, and 26.5% for $\text{Ti}_3\text{C}_2\text{T}_x\text{-25}$, $\text{Ti}_3\text{C}_2\text{T}_x\text{-150}$, and $\text{Ti}_3\text{C}_2\text{T}_x\text{-250}$, respectively. In addition, the mass activities of $\text{Ti}_3\text{C}_2\text{T}_x\text{-25}$, $\text{Ti}_3\text{C}_2\text{T}_x\text{-150}$, and $\text{Ti}_3\text{C}_2\text{T}_x\text{-250}$ were 50, 104, and 365 $\text{mmol}\cdot\text{g}^{-1}\cdot\text{h}^{-1}$, respectively. As

shown in Supplementary Fig. 10, a linear correlation was obtained between the percentage of O-Ti species and the activity of HCOOH dehydrogenation. Thus, the activity of HCOOH dehydrogenation for $\text{Ti}_3\text{C}_2\text{T}_x$ MXenes was highly related to the amount of O-Ti species on the surface of $\text{Ti}_3\text{C}_2\text{T}_x$ MXenes. We have added some sentences, one figure and one table in the revised manuscript and supplementary information (p. 6, lines 2-6; p. 7, lines 23-26; p. 15, lines 22-27; Supplementary Fig. 10; Supplementary Table 3; highlighted in yellow color).

“3). They use IR spectroscopy and say that they observe the bands of formate species at 2882, 1552, 1455 cm^{-1} in 6 spectra for 3 samples. I have experience in IR spectroscopy and do not see reliably that they have these 3 lines in their spectra. Additional confirmation could be a study of the OH groups region. Some hydroxyls could interact with formic acid and this could be observed in IR spectra. However, the authors do not show this region for the in situ studies.”

Thanks for the reviewer’s suggestion. In the original manuscript, *in situ* DRIFT spectra were collected from 900 to 3200 cm^{-1} . The peak of OH group is generally located at 3000-3700 cm^{-1} . To further elaborate the information of the OH group region, we have re-collected *in situ* DRIFT spectroscopy for HCOOH dehydrogenation over $\text{Ti}_3\text{C}_2\text{T}_x$ -250 from 900 to 4000 cm^{-1} . *In situ* DRIFT experiments were conducted in an elevated-pressure cell (Harrick DRK-4-BR4) with a Fourier transform infrared spectrometer (Bruker TENSOR II). After treatment of the catalysts with a flowing of N_2 (1 bar) for 20 min at 25 °C, the background spectrum was acquired. Then 1 bar of N_2 was allowed to bubble in HCOOH solution and flowed into the cell for 20 min at 25 °C, followed by *in situ* DRIFT measurements, which was regarded as the *in situ* DRIFT spectrum at 0 min. Then, the cell was sealed and heated to 80 °C within ca. 20 s. *In situ* DRIFT spectra were then recorded at interval at 80 °C. As for $\text{Ti}_3\text{C}_2\text{T}_x$ -250 shown in Fig. 5a, the bands of HCOO* species at 2882, 1555, 1455 cm^{-1} were obviously observed. The frequencies at 2945 cm^{-1} , 1790 cm^{-1} , 1208 cm^{-1} , and 1100 cm^{-1} were assigned to the stretching vibration of C-H, the stretching vibration of C=O, the coupled stretching vibration of C-O with the deformation vibration of O-H, and the stretching vibration of C-OH in HCOOH* species. The absorption peaks ranging from 3000 to 3700 cm^{-1} were assigned to the $\nu(\text{O-H})$ stretching mode of surface OH groups. After reaction at 80 °C for 5 min, the peaks for HCOOH* species and HCOO* species significantly attenuated (Fig. 5a). More importantly, the peaks for physisorbed CO_2^* species at 2334 cm^{-1} and 2364 cm^{-1} strengthened tremendously. Further prolonging the reaction time to 10 min and 20 min, the peaks for HCOOH* species and HCOO* species were vanished. At the same time, the peaks for physisorbed CO_2^* species became the domination. In comparison, we have also re-collected the *in situ* DRIFT spectroscopy for HCOOH dehydrogenation over $\text{Ti}_3\text{C}_2\text{T}_x$ -25 under the same conditions. As shown in Figure 5b, the peaks for HCOOH* species slightly weakened while the peaks for HCOO* species remained almost unchanged. In addition, no obvious enhancement in peaks for physisorbed CO_2^* species was observed for $\text{Ti}_3\text{C}_2\text{T}_x$ -25. Thus, we assumed that HCOOH* could be consumed swiftly while the conversion of HCOO* to CO_2^* was blocked for $\text{Ti}_3\text{C}_2\text{T}_x$ -25.

As for the role of OH groups in catalytic reaction, XPS spectra and DRIFT spectra proved the

presence of OH groups on the surface of $\text{Ti}_3\text{C}_2\text{T}_x$ MXenes (Fig. 2a and Supplementary Fig. 6). More importantly, the H atom of HCOOH^* was able to dissociate to form HCOO^* and OH^* on the surface of $\text{Ti}_3\text{C}_2\text{T}_x$ MXenes. Thus, the absorption peak ranging from 3000 to 3700 cm^{-1} was obvious observed for *in situ* DRIFT measurements for both $\text{Ti}_3\text{C}_2\text{T}_x$ -25 and $\text{Ti}_3\text{C}_2\text{T}_x$ -250 (Figs. 5a-b). As for $\text{Ti}_3\text{C}_2\text{T}_x$ -25, the peak of OH^* species slightly weakened while the peaks of HCOO^* species remained almost unchanged from 0 to 20 min at 80 °C. With regard to $\text{Ti}_3\text{C}_2\text{T}_x$ -250, the peak of OH^* species was vanished from 0 to 20 min at 80 °C with the increasing of the peaks of CO_2^* species, because surface OH^* and hydrogen atom from HCOO^* combined to get H_2 and CO_2 . Thus, surface OH^* groups were nearly unable to promote the dehydrogenation of HCOOH . In addition, we also investigated the effect of OH^* species on HCOOH dehydrogenation from the theoretical perspective by putting a single H atom to pre-adsorb on $\text{Ti}_3\text{C}_2\text{O}_2$ surface ($\text{Ti}_3\text{C}_2\text{O}_2\text{-H}_1$) as a model (Supplementary Fig. 18a). As shown in Supplementary Figure 18b, $\text{Ti}_3\text{C}_2\text{O}_2\text{-H}_1$ exhibited a stronger adsorption for HCOOH with an adsorption energy of -0.53 eV than that of $\text{Ti}_3\text{C}_2\text{O}_2$ with the adsorption energy of -0.35 eV. In addition, we also simulated the process of H_2 formation on $\text{Ti}_3\text{C}_2\text{O}_2\text{-H}_1$, where the activation energy was computed to be 0.57 eV, very close to that over $\text{Ti}_3\text{C}_2\text{O}_2$ with the value of 0.61 eV (Supplementary Fig. 18c). Thus, surface OH groups on $\text{Ti}_3\text{C}_2\text{O}_2$ contributed little to the dehydrogenation of HCOOH . We have added some sentences and two figures in the revised manuscript (p. 9, lines 25-31; p. 10, lines 7-30; p. 16, lines 24-31; Fig. 5a-b; highlighted in yellow color).

“4). Also, if they calculate E_a of some steps by DFT can they compare with experimental values of the apparent E_a .”

In DFT calculations, ideal models are used for a certain catalytic system by searching catalytic process with close geometrical or/and electronic structures. Generally speaking, DFT calculations are powerful methods to explain and even predict the trends of catalytic performance, without directly correlation of calculated E_a values and experimental values of the apparent E_a (*Science* **2005**, 307, 555-558; *PNAS* **2011**, 108, 937-943). In this case, surface oxygen species were proved as the key parameters to govern catalytic activity for $\text{Ti}_3\text{C}_2\text{T}_x$ MXenes towards HCOOH dehydrogenation, well consistent with the experimental results.

“5). Hence, they need to prove further the presence of formate as this study is a key point for the establishment of the mechanism. They also have to provide more evidence for the dependence of the activity on the specific oxygen species.”

We greatly thank this reviewer for his/her suggestions. The results of *in situ* DRIFT and DFT calculations have confirmed that HCOO^* species was a key intermediate for HCOOH dehydrogenation. Besides, we have also performed the experiment of isotope labelling for dehydrogenation of formic acid over $\text{Ti}_3\text{C}_2\text{T}_x$ -250. As for formic acid, the hydrogen in the carboxyl group is able to be exchanged with deuterium oxide at room temperature, while the hydrogen in the methyl group is nearly unable to be exchanged with deuterium oxide under the

reaction conditions (*J. Chem. Phys.* **1957**, 27 (6), 1305-1308; *The Course of Formic Acid Reduction of Enamines* **1957**, 6210-6214; *J. Chem. Soc.* **1952**, 2125-2127). Thus, to investigate the kinetic isotopic effect (KIE), we have performed the dehydrogenation of formic acid over $\text{Ti}_3\text{C}_2\text{T}_x$ -250 by employing different substrates including HCOOH, DCOOH, HCOOD, and DCOOD, and solvents including H_2O and D_2O . The experiment by using specific formic acid and water as the substrate and solvent was abbreviated as “formic acid+water”. In this case, we have implemented the following experiments: HCOOH+ H_2O , DCOOH+ H_2O , HCOOD+ D_2O , and DCOOD+ D_2O . As the rate of formic acid dehydrogenation is generally constant at the beginning of the reaction, the initial rate is used in KIE studies for formic acid dehydrogenation (*Nature Commun.* **2016**, 7, 11308). In the revised manuscript, the initial rate was named as $k_{(\text{formic acid}+\text{water})}$ by using specific formic acid and water as the substrate and solvent, respectively. The reaction profiles *versus* reaction time were shown in Fig. 4a, where gases were all gradually generated at different reaction conditions. The values of $k_{(\text{HCOOH}+\text{H}_2\text{O})}/k_{(\text{DCOOH}+\text{H}_2\text{O})}$, $k_{(\text{HCOOD}+\text{D}_2\text{O})}/k_{(\text{DCOOD}+\text{D}_2\text{O})}$, $k_{(\text{HCOOH}+\text{H}_2\text{O})}/k_{(\text{HCOOD}+\text{D}_2\text{O})}$, $k_{(\text{DCOOH}+\text{H}_2\text{O})}/k_{(\text{DCOOD}+\text{D}_2\text{O})}$, and $k_{(\text{HCOOH}+\text{H}_2\text{O})}/k_{(\text{DCOOD}+\text{D}_2\text{O})}$ were calculated to be 2.64, 3.61, 1.38, 1.41, and 4.41, respectively (Fig. 4b). Generally speaking, a first-order KIE is obtained when the value of $k_{\text{H}}/k_{\text{D}}$ was greater than ~ 1.5 (*ACS Catal.* **2017**, 7, 3850-3859; *Angew. Chem., Int. Ed.* **2012**, 51, 3066-3072). Thus, the first-order KIE was attributed to the dissociation of H/DCOO* to form COO* over $\text{Ti}_3\text{C}_2\text{T}_x$ -250, which was regarded as the rate-determining step. Moreover, the released gas has also been analyzed by mass spectroscopy. As shown in Fig. 4c and Fig. 4d, H_2 and D_2 were detected as the sole product in the cases of HCOOH+ H_2O and DCOOD+ D_2O , respectively. As for the cases of DCOOH+ H_2O and HCOOD+ D_2O , HD was detected as the dominated product, as well as a small amount of H_2 and D_2 , respectively (Figs. 4e-f). Accordingly, surface hydride and hydrogen atom from HCOO* combined to get H_2 . A nearly mono-molecular concerted mechanism rather than β -hydride elimination mechanism was regarded to be occurred over $\text{Ti}_3\text{C}_2\text{T}_x$ -250 towards the dehydrogenation of formic acid. These results were well consistent with the catalytic mechanism provided by *in situ* DRIFT and DFT studies (Fig. 5).

In addition, in order to elaborate the dependence of the activity on the specific oxygen species, we have also directly correlated the amount of surface O-Ti species to catalytic activity. As shown in Supplementary Fig. 10, a linear correlation was obtained between the percentage of O-Ti species and the activity of HCOOH dehydrogenation. Thus, the activity of HCOOH dehydrogenation for $\text{Ti}_3\text{C}_2\text{T}_x$ MXenes was highly related to the amount of O-Ti species on the surface of $\text{Ti}_3\text{C}_2\text{T}_x$ MXenes. We have added some sentences as well as two figures, one table and six references in the revised manuscript and supplementary information (p. 6, lines 2-6; p. 7, lines 23-26; p. 8, lines 28-31; p. 9, lines 1-31; p. 10, lines 1-2, line 5, lines 7-30; p. 13, lines 22-24; p. 15, lines 22-27; p. 16, lines 6-31; p. 21, lines 27-31; p. 22, lines 1-8; Figure 4; Supplementary Fig. 10; Supplementary Table 3; highlighted in yellow color).

REVIEWERS' COMMENTS:

Reviewer #1 (Remarks to the Author):

The authors responded all my comments satisfactorily apart from one point about the counter ion effect, and the response is totally unacceptable. Probably the reviewer did not make this point clear enough or the authors did not understand this point correctly. From the experimental point of view (preparation procedure), it is not possible to have atomic H, Al and F on the surfaces. For H and Al, their right form should be their positively charged species, like H(+) and Al(3+), while for F, the right form should be the negatively charged species, like F(-). For keep the charge balanced, they need the counter ions, i.e.; anions to balance H(+) and Al(3+), and cation to balance F(-). The authors should check this possibility. Otherwise, the reaction becomes radical process

The response that "Considering that H, Al and F atoms might exist and adsorb on the surface of catalysts" is just a joke and this does not reflect the chemical reality. Please delete this before making any joke

Reviewer #4 (Remarks to the Author):

The manuscript contains significant novel results. The authors have performed a serious work to change their manuscript. I am happy with the additional changes and support publication of this manuscript.

Point-by-point response to reviewer comments

Manuscript ID: NCOMMS-19-38445A

Reviewer #1 (Remarks to the Author):

“The authors responded all my comments satisfactorily apart from one point about the counter ion effect, and the response is totally unacceptable. Probably the reviewer did not make this point clear enough or the authors did not understand this point correctly. From the experimental point of view (preparation procedure), it is not possible to have atomic H, Al and F on the surfaces. For H and Al, their right form should be their positively charged species, like H(+) and Al(3+), while for F, the right form should be the negatively charged species, like F(-). For keep the charge balanced, they need the counter ions, i.e.; anions to balance H(+) and Al(3+), and cation to balance F(-). The authors should check this possibility. Otherwise, the reaction becomes radical process. The response that “Considering that H, Al and F atoms might exist and adsorb on the surface of catalysts” is just a joke and this does not reflect the chemical reality. Please delete this before making any joke.”

Thanks for the reviewer’s suggestion. Sorry for our misunderstanding about the reviewer’s point. We have investigated the promotion effect of H, Al and F species on the reaction by considering single H, Al and F atoms adsorbed on these catalysts as new catalysts theoretically. Once these single atoms are adsorbed on the catalysts, electron transfer between single atoms and their supports occurs. In this case, we have calculated the Bader charge of $\text{Ti}_3\text{C}_2\text{O}_2\text{-H}_1$ (neutral-charged model), where adsorbed H atom donates 0.63 e to $\text{Ti}_3\text{C}_2\text{O}_2$. Therefore, based on the proposed model, the reactions are not radical process. In addition, we have also computed the Bader charge distribution of the model by removing one electron from the $\text{Ti}_3\text{C}_2\text{O}_2\text{-H}_1$ system (positive-charged model), and found that the H species donates 0.64 e to the support. Relative to neutral-charged model, positive-charged model exhibits almost no extra electron accumulation/depletion of the H species. We have deleted the expression of “Considering that H, Al and F atoms might exist and adsorb on the surface of catalyst” in main text.

Reviewer #4 (Remarks to the Author):

The manuscript contains significant novel results. The authors have performed a serious work to change their manuscript. I am happy with the additional changes and support publication of this manuscript.

We sincerely appreciate the reviewer’s recognition of our work.